# CSF-1R in Cancer: More than a Myeloid Cell Receptor

**DOI:** 10.3390/cancers16020282

**Published:** 2024-01-09

**Authors:** Francesca Cersosimo, Silvia Lonardi, Cristina Ulivieri, Paolo Martini, Andrea Morrione, William Vermi, Antonio Giordano, Emanuele Giurisato

**Affiliations:** 1Department of Biotechnology Chemistry and Pharmacy, University of Siena, 53100 Siena, Italy; francesca.cersosi@student.unisi.it; 2Department of Molecular and Translational Medicine, University of Brescia, 25100 Brescia, Italy; silvia.lonardi@unibs.it (S.L.); paolo.martini@unibs.it (P.M.); william.vermi@unibs.it (W.V.); 3Department of Life Sciences, University of Siena, 53100 Siena, Italy; cristina.ulivieri@unisi.it; 4Center for Biotechnology, Department of Biology, Sbarro Institute for Cancer Research and Molecular Medicine, College of Science and Technology, Temple University, Philadelphia, PA 19122, USA; andrea.morrione@temple.edu; 5Department of Medical Biotechnology, University of Siena, 53100 Siena, Italy; giordano12@unisi.it

**Keywords:** CSF-1R, cancer cell signaling, cell proliferation, stemness, cell migration, chemoresistance

## Abstract

**Simple Summary:**

Several studies have highlighted the importance of the myeloid receptor CSF-1R in the context of tumors as a key player in the generation of an immunosuppressive microenvironment. Since recent research has demonstrated its expression also on the surface of cancer cells, the relevance of CSF-1R in this field has increased. Why might myeloid receptors be expressed by tumor cells? What advantages does CSF-1R expression provide to cancer cells? The aim of this review is to gather available data on CSF-1R expression in cancer cells to provide a new way to consider this receptor. Although previous works demonstrated the pro-tumoral role of CSF-1R in cancer cells in different tumor types, the precise mechanisms regulating its expression in cancer cells are still unknown and need further investigation in order to identify novel tumoral markers and the possible candidate for therapeutic intervention.

**Abstract:**

Colony-stimulating factor 1 receptor (CFS-1R) is a myeloid receptor with a crucial role in monocyte survival and differentiation. Its overexpression is associated with aggressive tumors characterized by an immunosuppressive microenvironment and poor prognosis. CSF-1R ligands, IL-34 and M-CSF, are produced by many cells in the tumor microenvironment (TME), suggesting a key role for the receptor in the crosstalk between tumor, immune and stromal cells in the TME. Recently, CSF-1R expression was reported in the cell membrane of the cancer cells of different solid tumors, capturing the interest of various research groups interested in investigating the role of this receptor in non-myeloid cells. This review summarizes the current data available on the expression and activity of CSF-1R in different tumor types. Notably, CSF-1R^+^ cancer cells have been shown to produce CSF-1R ligands, indicating that CSF-1R signaling is positively regulated in an autocrine manner in cancer cells. Recent research demonstrated that CSF-1R signaling enhances cell transformation by supporting tumor cell proliferation, invasion, stemness and drug resistance. In addition, this review covers recent therapeutic strategies, including monoclonal antibodies and small-molecule inhibitors, targeting the CSF-1R and designed to block the pro-oncogenic role of CSF-1R in cancer cells.

## 1. Introduction

Colony-stimulating Factor 1 Receptor (CSF-1R) is a tyrosine kinase receptor expressed primarily on the surface of monocytes and tissue macrophages, where it plays a central role in the regulation of immune responses and cell differentiation [1]. The receptor is activated via the binding of its cognate ligands, which induces receptor dimerization, autophosphorylation and the activation of downstream signaling pathways. CSF-1/M-CSF and IL-34 are the known ligands of CSF-1R and bind different regions in the Ig-like domain of the extracellular region of CSF-1R [2,3]. This receptor–ligand interaction is critical for immune development and function, as well as tissue homeostasis [4]. The dysregulation of CSF-1R expression was associated with the development of an immunosuppressive TME in different cancer types [5], and CSF-1R favored TME interactions and the recruitment of tumor-associated macrophages (TAMs) at cancer sites [6,7,8]. In human immune cells, genetic mutations in the extracellular region of CSF-1R induce tumorigenesis and constitutive receptor activation [9]. Additionally, mutations in the *c-fms* codon 301, leading to constitutive tyrosine kinase activity associated with neoplastic transformation [10]. Additionally, CSF-1R expression was found in the cancer cell membrane of different tumor types, where it is associated with the main cancer hallmarks including increased proliferation, migration and drug resistance [11,12,13,14]. The functional consequences of CSF-1R expression in tumor cells make it an interesting target for anti-cancer therapy. Indeed, CSF-1R induction induced a high proliferative index and anti-apoptotic signaling in breast and cervical cancer cells [11,14]. In other tumors, including mesothelioma, CSF-1R^+^ cells exhibited chemoresistance and expressed pluripotency-related genes [12]. Similarly, an altered expression of CSF-1R was found in resistant melanoma cells with BRAF and MAPKs mutations [13]. In this review, we summarize the evidence of CSF-1R expression in cancer tissue and cancer cell lines. Moreover, we described the pro-tumoral activity of CSF-1R in tumor cells in depth and the mechanisms leading to its activation. The therapeutic strategies targeting CSF-1R are also discussed; however, it should be noted that the primary focus of these studies was on CSF-1R-expressing TAMs. The expression of CSF-1R in cancer cells has only recently been studied, and there is still a lot to learn regarding this research area. However, as will be discussed later, evidence indicated that CSF-1R is an important factor in the development of tumor cell aggressiveness. Therefore, additional studies on different tumor types are required to determine whether CSF-1R is similarly involved in all malignancies and emphasize the significance of this receptor in the TME crosstalk, as well as cancer cell signaling. 

## 2. CSF-1R in Cancer Cells

In cancer, prolonged exposure to CSF-1R ligands leads to the recruitment of macrophages at the tumor sites and their differentiation into pro-tumorigenic macrophages, indicated as M2-like tumor-associated macrophages (M2-TAMs) [15]. Indeed, a correlation between increased circulating CSF-1 and enhanced CSF-1R^+^ TAMs was observed in breast cancer [16]. The development of an invasive phenotype in breast, ovarian and endometrial tumors was also associated with elevated levels of CSF-1R and circulating CSF-1 [17]. Similarly, gene expression profiling revealed increased *Csf-1* and *Csf-1r* mRNA expression levels in leiomyosarcoma (LMS) patient-derived samples. According to this study, LMS tumor cells secreted CSF-1 that, in turn, induced macrophage recruitment at the tumor sites. Additionally, in situ hybridization analysis in a set of LMS cases demonstrated that LMS cells produced both CSF-1 and CSF-1R, while macrophages did not secrete CSF-1, indicating an autocrine receptor activation at the tumor cell membrane [18]. The same mechanism was observed in ovarian cancer, where the blockade of this autocrine loop reversed the malignant phenotype [19]. Ide et al. reported the involvement of CSF-1R in prostate cancer carcinogenesis [20]. According to the study, low levels of CSF-1R and CSF-1 mRNA were detected in human prostate cancer cell lines, indicating the autocrine activation of the receptor. Further analysis indicated the expression of the receptor in prostatic intraepithelial neoplasia (PIN) and metastatic sites, suggesting the role of CSF-1R in prostate tumor development [20]. IHC analysis of prostatic adenocarcinoma indicated that CSF-1R expression was higher in metastatic tissue compared to non-metastatic controls, and the receptor was expressed by both cancer and stromal cells [21]. High levels of serum CSF-1 were considered the cancer biomarkers in certain tumors, including squamous cell carcinoma of the head and neck and colorectal cancer [22,23]. Li et al. described new roles of CSF-1R in hTERT immortal human epithelial ovarian cell lines. According to this study, CSF-1R was up-regulated and involved in the survival and growth of these cell lines. Additionally, CSF-1R represented an important component for hTERT cellular localization, as, in fact, CSF-1 stimulation induced the phosphorylation of NFkBp65 and the formation of a NFkBp65–hTERT complex. On the contrary, the genetic silencing of CSF-1R reduced nuclear levels of hTERT and complex formation. Lastly, they showed that CSF-1R activation regulated the transcription of hTERT via c-Myc activity [24].

CSF-1R mRNA is expressed in different cancer cell lines (Figure 1), and various studies reported that CSF-1R mRNA up-regulation correlated with poor prognosis in tumors [25,26]. Furthermore, oncogenic mutations in *c-fms* correlated with a worse prognosis in myelodysplastic syndrome [27]. The up-regulation of *c-fms* in cervical cancer cells was associated with the activation of TGF-β receptor signaling, and, in turn, TGF-1 receptor activation enhanced *c-fms* mRNA levels [28]. Barbetti et al. reported the CSF-1-dependent nuclear localization of CSF-1R in breast cancer cell lines for the first time, promoting the transcription of different proliferative genes such as *CCND1*, *c-JUN*, and *c-MYC* [29]. Soares et al. demonstrated high CSF-1R expression in renal carcinoma compared to normal tissues. FISH analysis indicated a copy number gain of CSF-1R in 59% of clear cell renal cell carcinoma samples. Moreover, three genetic changes in CSF-1R sequence, including mutations in exon 7 and exon 22, were identified in clear renal cell cancer [30]. Previous work demonstrated CSF-1R expression at the membrane and in the nucleus of CSF-1-stimulated cervical, ovarian and breast cancer cell lines [31]. Epigenetic alterations in the CSF-1R promoter region induced CSF-1R up-regulation in melanoma cells with the BRAF mutation. Interestingly, these cancer cells did not express the transcriptional factor PU.1, a key regulator of CSF-1R expression in myeloid cells, thereby suggesting that CSF-1R expression is regulated by different transcriptional mechanisms in melanoma cells [13]. An earlier study demonstrated that CSF-1R^+^ breast cancer cells were associated with a worse prognosis and tumor spread, mostly in ER^+^ breast cancer [26]. In pancreatic cancer, tumor cells expressed various CSF-1R levels, which were high in nerve-invasive PCCs, indicating that CSF-1R expression correlated with perineural migration and a worse prognosis [32].

### 2.1. Role of CSF-1R Ligands in TME Crosstalk

The communication between the tumor, stromal and immune cells in the tumor microenvironment (TME) is necessary for the establishment of an environment that is suitable for tumor cell growth [33], proliferation and invasion of secondary tissues [34]. In this scenario, CSF-1R ligands, IL-34 and M-CSF, play an important role in the crosstalk between immune and tumor cells, as, in fact, immune and cancer cells secreted cytokines and interleukins, which supported tumor development and progression [33,35]. In colorectal cancer (CRC), cancer cells secrete IL-34 and express CSF-1R, suggesting the presence of an autocrine positive feedback loop on the receptor. Moreover, IL-34-dependent CSF-1R activation increased the proliferative index of tumor cells [36]. In ovarian cancer, the production of IL-34 by tumor cells was associated with the development of a malignant microenvironment and tumor progression [37]. According to Shao-Lai Z. [38], IL-34, produced by hepatocellular carcinoma (HCC) cells, induced the recruitment of TAMs at tumor sites. In vivo assays showed that increased IL-34-related TAM infiltration was correlated with metastasis and poor prognosis in HCC [38]. In addition, in malignant pleural mesothelioma (MPM), high levels of IL-34 were detectable in MPM patients. The same research group demonstrated that tumoral IL-34 strongly supported the development of immunosuppressive phenotypes [39]. Recently, it was discovered that IL-34 stimulated CRC cell proliferation and migration through the activation of the ERK1/2 signaling pathway [36,40]. Additional studies demonstrated the involvement of the MEK/ERK pathway and PI-3K signaling in CSF-1-mediated proliferation, invasion and survival of lung cancer cell lines [41]. An IL-34- and M-CSF-enriched TME was associated with aggressive tumors and low overall survival in a group of lung cancer patients [42]. In endometrial cancer, tumor cells drove macrophage infiltration via CSF-1 secretion, and CSF-1 silencing reduced the migratory ability of macrophages [43]. Kirma et al. demonstrated that cervical cancer cells produced high levels of CSF-1, and CSF-1R blockade reduced the migratory capacity of cancer cells, suggesting the involvement of CSF-1 in tumor malignancy via the autocrine receptor activation [28].

### 2.2. CSF-1R in Cancer Cell Proliferation

Table 1 shows the expression in a tumor cell line explored by the Kaciniski group [17]. This work demonstrated the expression of CSF-1R transcripts in endometrial, ovarian and breast cancer cells. In addition, they discovered that glucocorticoids and lactogenic steroid enhanced CSF-1R expression at messenger RNA and protein levels in mammary epithelial cells [17]. In a CSF-1R-overexpressing MCF-7 cancer cell line, CSF-1 induced cell cycle arrest associated with increased p21 levels and the formation of p21/CDK complexes preventing cell cycle progression. In contrast, in T47-D cells, CSF-1 had pro-mitogenic functions and induced low levels of p21, suggesting the cell background-dependent role of CSF-1R [44]. Notably, another study conducted in metastatic breast cancer cells showed that in vivo CSF-1R up-regulation was mediated by TGF-β levels in the microenvironment, as in fact, the CSF-1R levels decreased upon TGF-β inhibition [45]. The authors also demonstrated that claudin-low cells expressed higher CSF-1R levels than luminal ones. In contrast to reports showing that CSF-1R suppression increased the proliferation rate of claudin-low cancer cells in vivo [46], Rovida et al. demonstrated that the inhibition of the receptor and autocrine M-CSF reduced the cell growth of claudin-low breast cancer cells [11]. Additionally, the proliferative action of CSF-1R correlated with the activation of ERK1/2, c-Jun, Cyclin D, and c-Myc downstream signaling pathways (Figure 2) [11]. In vitro and in vivo studies indicated that IL-34/CSF-1R activation increased BrdU incorporation, colony forming abilities of breast cancer cells and tumor growth in mouse models. Moreover, the tumorigenic role of IL-34 is mediated by the CSF-1R-dependent activation of MEK/ERK and JNK/c-Jun pathways [47]. This study also demonstrated the involvement of PIN1 in the IL-34-mediated phosphorylation of MEK, ERK, JNK and c-Jun [47], while the pro-tumorigenic role of CSF-1R was mediated by its interaction with the transforming growth factor-β-stimulated clone-22 (TSC-22) protein in cervical cancer cells. TSC-22 acts as onco-suppressor, and it is expressed at low levels in tumors, including cervical cancer. CSF-1R was found as a target of TSC-22 that interacted with the tyrosine domain of its intracellular region by blocking its activity [14]. In melanoma cell lines, the pharmacological and genetic targeting of the receptor inhibited proliferation of 3D cultures and increased apoptotic rate, suggesting the pro-survival role of CSF-1R [13]. CSF-1R-overexpressing glioma cells showed increased cell viability, ki-67-positivity and enhanced colony forming ability [48]. Interestingly, CSF-1R suppression affected cell cycle progression by enhancing p27 expression, thereby preventing cells entering the S phase. Moreover, CSF-1R overexpression did not induce AKT activation, but only ERK1/2 activity in glioma cells [48]. On the contrary, in T-cell lymphoma cells, CSF-1R activation led to AKT phosphorylation in a PI3K-dependent manner [49]. Rattanaburee et al. identified CSF-1R as a possible target of kusunokinin, a lignan-derived molecule with demonstrated anti-cancer efficacy. They demonstrated that kusunokinin down-regulated CSF-1R expression levels by binding to the juxta-membrane domain (JMD) of the receptor. The anti-proliferative activity of kusunokinin observed in breast cancer cells was associated with the suppression of the CSF-1R and AKT pathways. Kusunokinin-mediated CSF-1R suppression also decreased the expression levels of G2-M markers, including Cyclin B, CDK1 and c-Myc [50]. In canine mammary cancer cells, CSF-1R inhibition increased apoptosis and decreased Ki-67 positivity, suggesting that CSF-1R supported the survival and growth of cancer cells [51] (Table 1).

### 2.3. CSF-1R in Cancer Cell Migration

Cancer cell migration is a complex process involving different molecular components and evidence supports the role of CSF-1R in this process [52,57]. Sapi et al. showed that mammary epithelial cell invasiveness was associated with CSF-1R expression, and the invasive capacity of these cells was inhibited by a dominant negative mutant of the transcriptional factor Ets2, suggesting the involvement of Ets2 in modulating CSF-1R signaling [58]. In mammary epithelial MCF-10A cells, the constitutive activation of CSF-1R led to the hyperproliferation and destruction of acinar structures. In addition, the expression of a Y561F CSF-1R mutant in normal human breast cells failed to activate Src family members, indicating that Src signaling is required for CSF-1R action in modulating acinar structure integrity [52]. Intriguingly, the autocrine activation of CSF-1R enhanced the intracellular localization of E-cadherin and early wound closure [52]. An in vivo study demonstrated that CSF-1R inhibition negatively affected the invasiveness potential of claudin-low breast cancer cells; a tumor derived from mice orthotopically injected with CSF-1R-depleted breast cancer cells showed decreased invasive ability and reduced lung metastases as compared to controls, thereby suggesting the crucial role of CSF-1R in cancer cell motility and invasion [46]. A partial epithelial-to-mesenchymal transition (EMT) characterized by the expression of both Vimentin and E-cadherin was observed in inflammatory breast cancer, where treatment with the CSF-1R inhibitor BLZ945 reduced the spindle-like phenotype of cancer cells and reversed the partial EMT, indicating the role of CSF-1R in breast cancer invasiveness [59]. In canine mammary tumor cells, a genetic blockade of CSF-1R negatively affected the migratory and invasive abilities of these cells [51]. The overexpression of CSF-1 in human ovarian cancer cell lines induced the acquisition of invasive and metastatic characteristics observed both in vitro and in vivo. Moreover, the metastatic activity of CSF-1 was mediated by the downstream activation of the urokinase-type plasminogen activator (uPA) pathway, which was linked to motility and invasiveness in several tumor types [19]. Similarly, the administration of CSF-1 enhanced the invasive potential of lung cancer both in vivo and in vitro. Data showed that injection of CSF-1-depleted cancer cells in mice reduced tumor mass and osteolytic bone metastases compared to controls, as well as the percentage of Ki-67-positive tumor cells [60]. In human osteosarcoma cell lines, a pool of CSF-1R^+^ cells was identified and receptor expression was associated with mesenchymal marker expression and invasive capabilities, as CSF-1R genetic ablation inhibited EMT and the migration of tumor cells. Interestingly, the study identified *JAG1* as a downstream target gene involved in CSF-1R-dependent cell migration [56]. Shi et al. identified the CSF-1R/STAT3/Mir-34a axis as regulator of pro-tumoral functions in CRC, where CSF-1R expression was positively correlated with the expression of EMT-related genes like *SNAIL* and *SLUG*, and EMT markers such as vimentin and low expression levels of E-cadherin (Figure 2) [54]. Similarly, CSF-1R knockdown in glioma cells showed a reduced expression of EMT markers such as vimentin [48]. A functional role of CSF-1R in melanoma spread was also demonstrated by Giricz et al., showing a dose-dependent decrease in melanoma invasiveness after treatment with the CSF-1R inhibitor PLX3397 [13]. The clinical importance of CSF-1R was reported in gastric cancer (GC) [61]. According to this study, CSF-1/CSF-1R expression correlated with an advanced stage of disease in young patients and tumor metastasis. RT-qPCR and immunohistochemistry indicated CSF-1R expression by GC cells. An in vitro assay demonstrated the involvement of the CSF-1/CSF-1R axis in tumor migration via anoikis resistance and the induction of pro-angiogenic factor expression, such as VEGFA, in GC tissues [61] (Table 2). 

### 2.4. CSF-1R in Drug Resistance and Stemness

Vemurafenib-resistant melanoma cancer cell lines showed aberrant CSF-1R expression and a positive link between the transcription factor RUNX1, CSF-1R and IL-34, which affects the transcriptional regulation of CSF-1R [13]. Notably, CSF-1R and RUNX1 inhibitors reduced the aggressiveness of tumor cells and colony size in melanoma [13]. CSF-1R expression was up-regulated in a 5-FU chemo-resistant population of CRC cells [54], and the acquisition of the resistant phenotype, as well as the expression of stem-like and EMT-related genes, was linked to the down-regulation of Mir-34a, thereby suggesting that the CSF-1R/mir-34a pathway could be a potential target against CRC chemoresistance and invasiveness [54]. Mir-34a deficiency correlated with CSF-1R up-regulation and increased expression of the stemness marker *Lgr5*, as well as tumoroid formation capabilities in intestinal adenoma cells [55]. Studies on mesothelioma primary cultures and cell lines showed that the autocrine activation of CSF-1R characterized a pool of malignant cells with stem-like and chemo-resistant phenotypes. The expression of the stemness markers *SOX2*, *NANOG*, *OCT4*, *c-MYC* and *CD44* was observed in CSF-1R^+^ cells in association with increased ALDH activity and *ABCG2* expression (Figure 2) [12], suggesting a link between CSF-1R expression and the acquisition of drug-resistant phenotypes. Chemoresistance of CSF-1R^+^ cells correlated with the activation of the AKT signaling pathway, as an AKT blockade sensitized cells to cell death following Pemetrexed treatment [12]. Similarly, CSF-1R overexpression induced chemoresistance in ovarian cancer cells via AKT and ERK1/2 pathway activation, while knockdown studies indicated that CSF-1R inhibition increased the apoptotic rate of cisplatin-treated tumor cells [53]. Pass et al. showed a population of Cisplatin-resistant CSF-1R^+^ lung cancer cell lines [62]. Furthermore, the group demonstrated that CSF-1R expression is necessary for chemoresistance in lung cancer. Indeed, the inhibition of the receptor with the tyrosine kinase (TK) inhibitor JNJ-40346527 impaired the sphere-forming capabilities and the expression of both stemness and chemoresistance markers in lung cancer cells. The study also provided evidence of the synergic activity of cisplatin after CSF-1R TK inhibition in vivo [62] (Table 3).

## 3. Anti-CSF-1R Therapeutic Strategies and Future Prospectives

Several CSF-1R inhibitors, including small molecules and neutralizing antibodies, have been developed over the last years, and many drugs targeting the CSF-1R are currently in clinical development or approved for tumor treatment for many tumor types [63,64]. Studies reported that the immunotherapeutic activity of anti-CSF-1R molecules due to CSF-1R expression in myeloid cells correlated with the development of an immunosuppressive TME [6,65]. However, few data are currently available on the efficacy of anti-CSF-1R drugs in cancer. Pexidartinib (PLX3397) was tested in phase I clinical trials in several solid tumors, either alone or in combination with Paclitaxel, demonstrating good tolerance and low toxicity [66,67]. PLX3397 was approved for the treatment of symptomatic tenosynovial giant cell tumor (TGCT), which works through depleting immunosuppressive cells [68]. This FDA-approved CSF-1R inhibitor also interferes with the growth and survival of T cell lymphoma cells by acting on both tumoral and microenvironmental cell pools [49]. In BRAF-therapy-resistant melanoma, combinatorial treatment with CSF-1R (PLX3397) and BRAF V600E (PLX4720) inhibitors at a low dose strongly increased the survival rate of murine xenograft models, as well as decreased the proliferation and invasiveness of cancer cell lines [13]. Previous studies already demonstrated the efficacy of the combinatorial strategy in reducing tumor-infiltrating myeloid cells in a syngeneic mouse model of *BRAF^V600E^*-driven melanoma [69]. The efficacy of Pexidartinib was also demonstrated by reducing the proliferation of CSF1R-positive TGCT cell lines. The combined use of Pexidartinib and Sotuletinib (BLZ945), another CSF-1R inhibitor, reduced the proliferation of tumor cells, their spheroid formation ability, as well induced apoptosis in correlation with CSF-1R levels in TGCT [70]. Additionally, PLX3397 administration effectively decreased osteosarcoma tumor growth and the metastatic potential of tumor cell line in patient-derived xenograft (PDX)-injected animal models [71]. Another CSF-1R kinase inhibitor, Imatinib, has showed a strong reduction in the colony forming ability of breast cancer cells [11]. This potent anti-tumor activity of anti-CSF-1R drugs has been mostly attributed to its action on TAMs and other myeloid-derived cells [63]. A thorough understanding of the significance of CSF-1R expression in tumor cells will greatly help in developing therapeutic drugs that specifically target the neoplastic component, for single use or in combination with other anti-cancer therapies. 

## 4. Conclusions

An aberrant expression of CSF-1R has been observed in tumors and it is associated with a worse prognosis, while high levels of CSF-1R are predictive of an immunosuppressive phenotype and aggressive tumors [5]. Recently, CSF-1R expression was also detected in the cell membrane of cancer cells, where it plays an important role in supporting cancer progression. Thus, the characterization of the mechanisms regulating the expression of a myeloid receptor in cancer cells is an intriguing topic for future studies. Notably, not all tumor cells expressed the receptor, but only a subset of cells with aggressive behavior, thereby indicating that CSF-1R could contribute to promoting tumor aggressiveness. As highlighted in this review, CSF-1R expression in tumor cells is indeed associated with increased proliferation, expression of EMT markers, development of invasive capabilities, drug resistance and acquisition of stem-like features. The major CSF-1R-mediated signaling pathways so far described in cancer cells are summarized in Figure 2. Although it was previously thought that anti-CSF-1R therapeutic strategies only targeted M2-TAMs, the evidence demonstrating CSF-1R expression in cancer cells and its contribution to tumor growth makes this receptor an interesting target for therapy. However, very little is still known regarding CSF-1R expression and downstream signaling mechanisms in cancer cells. The main issues in the study of CSF-1R expression and function in cancer cells include the fact that it is expressed at low levels compared to the expression in myeloid cells, making it difficult to detect in some experimental settings, such as tumor tissues, where monocytes/macrophages represent the majority of CSF-1R^+^ cells.

Different questions are opened regarding CSF-1R activity and regulation in cancer cells. Which cells express CSF-1R? Under which conditions? Are CSF-1R regulation mechanisms shared between cancer cells of different tumor types? It is possible to hypothesize that CSF-1R is positively expressed in a specific population of cells or that its expression is differently regulated during the cell’s life cycle. As mentioned above, many studies observed the involvement of CSF-1R signaling in the promotion of stemness and chemoresistance mechanisms, making CSF-1R in tumor cells a negative predictor of outcome. Additionally, its activity in cell cycle progression suggested a specific role and regulation in proliferation. Is CSF-1R differently regulated than other tumor cell membrane receptors? Finally, the possibility that CSF-1R exerts different functions based on its cellular localization is another point to deeply investigate. Understanding the receptor’s cellular localization may provide important indications for its activity and the mechanism regulating its function in tumor cells, as CSF-1R nuclear localization may suggest the transcriptional activity of the receptor. To conclude, exploring the molecular mechanisms modulating CSF-1R expression and function in cancer cells and whether tumor cells share the same regulatory mechanisms of immune cells will greatly help design novel approaches to target CSF-1R in cancer. 

## Figures and Tables

**Figure 1 cancers-16-00282-f001:**
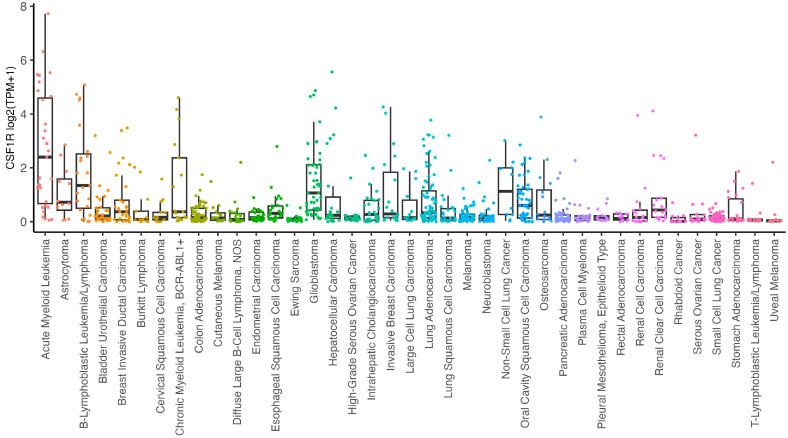
CSF1R mRNA expression (log2-transformed, using a pseudo-count of 1) in cell line models from the Dependency Map (DepMap) Public 23Q4 (https://depmap.org/portal/ (accessed on 28 December 2023)). Cell lines are grouped by cancer types defined as Oncotree Subtype. Cancer subtypes with more than 10 cell line models are shown.

**Figure 2 cancers-16-00282-f002:**
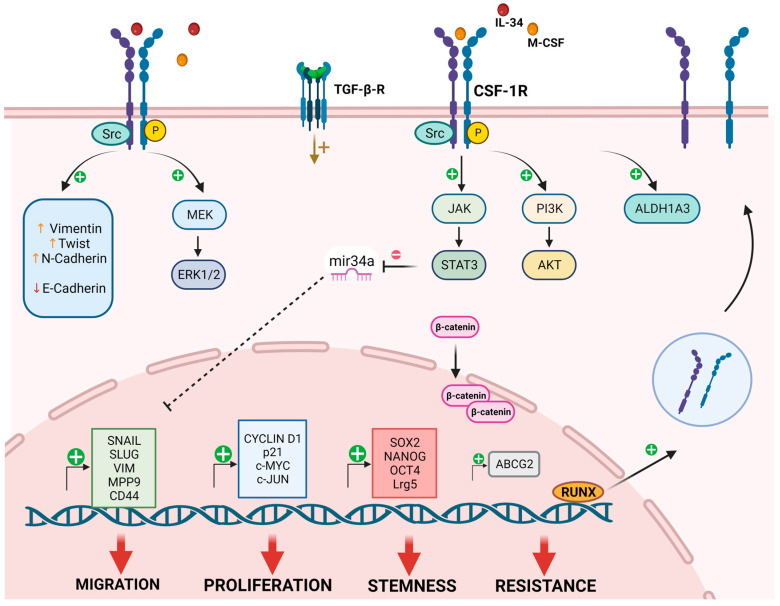
CSF-1R signaling in cancer cells. The image is a schematic representation of the available data regarding CSF-1R signaling and function in different cancer cell types. In mammary epithelial cells, Src signaling correlated with CSF-1R activation [52]. ERK 1/2 pathway activation following CSF-1R induction was reported in lung, breast and glioma cancer cells, associated with cell proliferation [11,41,48]. In breast cancer, CSF-1R signaling induced the transcription of the proliferative genes *CCND1*, *c-JUN* and *c-MYC* [29], even though MCF-7 cells overexpressing CSF-1R increased the expression levels of p21 [44]. In different tumor types, CSF-1R promoted proliferation and chemoresistance via the activation of PI3K/AKT signaling [12,49,53]. In colorectal cancer, the CSF-1R/STAT3/mir-34a axis was involved in the development of a chemoresistant and invasive phenotype [54]. Moreover, mir-34a negatively regulated the expression of the stem-like gene *Lrg5* and the acquisition of stem-like traits [55]. In mesothelioma, CSF-1R expression correlated with stemness marker expression, such as *SOX2*, *NANOG*, *OCT4* and *c-MYC*, and drug-resistance markers including *ABCG2* transcription [12]. In colorectal cancer, glioma and mesothelioma CSF-1R induced the expression of EMT-related genes, like *SNAIL* and *SLUG*, *Vimentin* (VIM) and *matrix metalloproteinase-9* (*MMP-9*) and *CD44* [12,48,54]. Moreover, in osteosarcoma cells, CSF-1R expression was associated with Twist and N-cadherin expression [56]. According to Giricz et al. [13], in a subset of chemoresistant melanoma cells, the main regulator of CSF-1R expression might be RUNX1, which is associated with the CSF-1R promoter region [13]. In both cervical and breast cancer, the expression levels of CSF-1R linked with TGF-β receptor activity [28,46]. Accumulation of nuclear beta-catenin was also regulated via CSF-1R in intestinal tumor cells [55]. Created in Biorender.com (accessed on December 22, 2023).

**Table 1 cancers-16-00282-t001:** CSF-1R in cancer cell proliferation.

	Proliferation	
Tumor Type	Function	Reference
Breast Cancer	Cell cycle regulation p21 induction	[44]
Breast Cancer	ERK1/2; c-Jun; Cyclin D; c-Myc activation	[11]
Breast Cancer	MEK/ERK; JNK/c-Jun c-Fos; c-Jun; AP-1 activation	[47]
Cervical Cancer	TSC-22-mediated proliferation	[14]
Melanoma	Pro-survival; anti-apoptotic	[13]
Glioma	p27 expression Increased cell viability Ki67^+^ cells ERK1/2 activation	[48]
T-cell Lymphoma	PI3-K/AKT activation	[49]
Breast Cancer	G2-M markers expression	[50]
Mammary Epithelial Tissue	Hyperproliferation	[52]

**Table 2 cancers-16-00282-t002:** CSF-1R in cancer cell migration.

	Migration	
Tumor Type	Function	Reference
Mammary Epithelial Tissue	Destruction of acinar architecture E-Cadherin re-localization Src-kinase activity	[52]
Breast Cancer	Invasiveness and metastases	[46]
Breast Cancer	EMT	[59]
Ovarian Cancer	Adhesion and motility Tumor metastases Increased urokinase surface expression	[19]
Lung Cancer	Osteolytic metastases	[60]
Osteosarcoma	EMT via JAG-1 pathway activation	[56]
Colorectal Cancer	EMT markers expression STAT3 induction Mir-34a down-regulation	[54]
Glioma	EMT	[48]
Melanoma	Invasiveness	[13]
Gastric Cancer	Cell migration and tumor metastasis	[61]

**Table 3 cancers-16-00282-t003:** CSF-1R in drug resistance and stemness.

	Drug Resistance and Stemness	
Tumor Type	Function	Reference
Melanoma	BRAF resistance	[13]
Colorectal Cancer	ALDH activity mir-34a down-regulation	[54]
Mesothelioma	ALDH activity Stem-like markers expression AKT pathway activation	[12]
Intestinal Adenoma	Stem-like gene expression	[55]
Ovarian Cancer	Cisplatin resistance AKT; ERK1/2 pathway activation	[53]
Lung Cancer	Cisplatin resistance Stem-like markers expression	[62]

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
