# Peer review of "CSF-1R in Cancer: More than a Myeloid Cell Receptor"

_cancers, 2024, doi:10.3390/cancers16020282_

Round 1
Reviewer 1 Report
Comments and Suggestions for Authors
The review "CSF 1R in Cancer: More than a Myeloid Cell Receptor" from Cersosimo et al. is a well-written and informative article contextualizing the importance of the myeloid receptor CSF-1R in several tumors. Considering that CSF1-R is canonically known as a tyrosine kinase receptor expressed on the surface of macrophages and microglia, this review article addresses the important topic of its observed roles in various cancers. The review starts with a comprehensive introduction, with properly cited seminal works, and proceeds with framing CSF-1R in cancer and considering its involvement in cancer cells, its ligands in the tumor microenvironment crosstalk, in the proliferation of the cancer cells, in cancer cell migration, in drug resistance and stemness. A final paragraph is dedicated to the therapeutic strategies landscape involving CSF1-R.
Overall, this is an interesting and well-written review article that does a very good job of contextualizing modern literature about CSF1-R. The tables and the figures are well-made and informative. The citations throughout the paper are well-selected, appropriate, and relevant, encompassing recent publications from the last 5 years as well.
There are very few and minor concerns about this article.
One would be that in the introduction when presenting CSF1-R and its ligands, my opinion is that the article would benefit from having a figure with schematics of these proteins and/or recent 3D structures. The authors could consider adding this.
In the introduction, at line 53, the authors mention the TAMs. Although a reader knowledgeable of the field would know that these are probably Tumor-Activated Macrophages, it would be beneficial to write this acronym in extended form before using its abbreviation, the first time it is mentioned (as it is done for all other abbreviations).
In paragraph 2.5, at lines 263 to 264, there is a sentence in italics. Was this intentional, or a mistake? Usually italic is reserved for sub-titles, genes, or other conventions, but not to convey emphasis.
Comments on the Quality of English LanguageAlso for the quality of the English Language, I have very few and minor things to point out. Generally, the text is well-written, precise, and clear. Some sentences throughout the article would sound better if conveyed differently.
For example, please mind:
Line 63, "there is still much to learn regarding the area" could benefit from being rewritten to sound better in English.
Author Response
Reviewer 1
Comments and Suggestions for Authors
The review "CSF 1R in Cancer: More than a Myeloid Cell Receptor" from Cersosimo et al. is a well-written and informative article contextualizing the importance of the myeloid receptor CSF-1R in several tumors. Considering that CSF1-R is canonically known as a tyrosine kinase receptor expressed on the surface of macrophages and microglia, this review article addresses the important topic of its observed roles in various cancers. The review starts with a comprehensive introduction, with properly cited seminal works, and proceeds with framing CSF-1R in cancer and considering its involvement in cancer cells, its ligands in the tumor microenvironment crosstalk, in the proliferation of the cancer cells, in cancer cell migration, in drug resistance and stemness. A final paragraph is dedicated to the therapeutic strategies landscape involving CSF1-R. Overall, this is an interesting and well-written review article that does a very good job of contextualizing modern literature about CSF1-R. The tables and the figures are well-made and informative. The citations throughout the paper are well-selected, appropriate, and relevant, encompassing recent publications from the last 5 years as well.
There are very few and minor concerns about this article.
One would be that in the introduction when presenting CSF1-R and its ligands, my opinion is that the article would benefit from having a figure with schematics of these proteins and/or recent 3D structures. The authors could consider adding this.
Response: We thank the reviewer for the positive comments. The structure of CSF1-R and its ligands have been already schematically represented in several manuscripts/reviews previously published (such as https://doi.org/10.1038/s12276-020-0484-z; https://doi.org/10.1016/j.str.2015.06.019 and others); thus, we decided that was not necessary to add a new one.
In the introduction, at line 53, the authors mention the TAMs. Although a reader knowledgeable of the field would know that these are probably Tumor-Activated Macrophages, it would be beneficial to write this acronym in extended form before using its abbreviation, the first time it is mentioned (as it is done for all other abbreviations).
Response: As required, at line 54, the acronym TAMs is fully spelled as Tumor-Associated Macrophages.
In paragraph 2.5, at lines 263 to 264, there is a sentence in italics. Was this intentional, or a mistake? Usually italic is reserved for sub-titles, genes, or other conventions, but not to convey emphasis.
Response: the sentence at lines 263-264 (lines 286-287 in the new version of the review) has been corrected.
Comments on the Quality of English Language
Also for the quality of the English Language, I have very few and minor things to point out. Generally, the text is well-written, precise, and clear. Some sentences throughout the article would sound better if conveyed differently.
For example, please mind:
Line 63, "there is still much to learn regarding the area" could benefit from being rewritten to sound better in English.
Response: All the text has been checked and some sentences have been rewritten.
Reviewer 2 Report
Comments and Suggestions for Authors
A review by Cersosimo et al. focuses on the role of CSF-1R in cancer. The topic of the paper is interesting and important. The manuscript is generally well written.
Specific comments:
1. Figure 1 is of unknown significance. First, this is a review paper. Second, it shows only expression of CSF-1R in few cell types. The Authors should consider to perform e.g., bioinformatic analysis of publicly available RNA-seq data and compare expression of CSF-1R more broadly.
2. The Authors should refer to more recent papers. Only the minority of papers that were mentioned have been published in the last 2-3 years.
3. Are ther any ongoing clinical trials assessing the activity of anti-CSF-1R drugs? If so, additional table showing these studies should be prepared.
Author Response
Reviewer 2
Comments and Suggestions for Authors
A review by Cersosimo et al. focuses on the role of CSF-1R in cancer. The topic of the paper is interesting and important. The manuscript is generally well written.
Specific comments:
- Figure 1 is of unknown significance. First, this is a review paper. Second, it shows only expression of CSF-1R in few cell types. The Authors should consider to perform e.g., bioinformatic analysis of publicly available RNA-seq data and compare expression of CSF-1R more broadly.
Response: as requested, Figure 1 has been replaced by a new revised Figure 1 showing CSF1R mRNA expression (log2 transformed, using a pseudo-count of 1) in cell line models from Dependency Map (DepMap) Public 23Q4 (https://depmap.org/portal/). Cell lines are grouped by cancer types defined as Oncotree Subtype. Cancer subtypes with more than 10 cell line models are shown.
- The Authors should refer to more recent papers. Only the minority of papers that were mentioned have been published in the last 2-3 years.
Response: To our knowledge all the references regarding CSF1R expression and function in cancer cells have been included in the References section. We selected appropriate, and relevant citations of CSF1R in cancer cells, also encompassing recent publications from the last 3-5 years.
- Are there any ongoing clinical trials assessing the activity of anti-CSF-1R drugs? If so, additional table showing these studies should be prepared.
Response: Clinical trials regarding specific anti-CSF1R drugs in cancer have been already published in several articles/reviews. However, to our knowledge, there are not ongoing clinical trials targeting CSF1R specifically for cancer cells.
Reviewer 3 Report
Comments and Suggestions for Authors
The manuscript provides a comprehensive overview of the current landscape of CSF1R inhibitors in cancer treatment. It effectively highlights the therapeutic potential of Pexidartinib (PLX3397) in various solid tumors and their impact on tumor growth and metastasis. The author emphasizes the need for further investigation into the molecular mechanisms regulating CSF1R expression in cancer cells and raises pertinent questions for future research. While generally well-structured and interesting to read, some comments for author consideration are listed below:
Comments for authors:
Title:
The title accurately reflects the content of the study and is concise and informative.
Introduction:
1. While the section discusses CSF-1R expression in cancer cells, it could benefit from further clarification on the functional consequences of this expression. Elaborating on how CSF-1R expression in cancer cells contributes to specific aspects of tumor biology would enhance the section's depth.
2. The addition of specific examples or case studies illustrating the impact of CSF-1R expression in different cancer types could provide a more concrete and relatable dimension to the discussion.
- In-text Citations: While the section makes several references to studies (e.g., "Src signaling correlated with CSF1R activation (52)"), I recommend the inclusion of more citations. Directly citing relevant studies would strengthen the section and provide readers with specific sources for further exploration.
- Mechanistic Details: The section could benefit from a more in-depth exploration of the molecular mechanisms underlying the effects of CSF1R in cancer cells. Including detailed insights into how CSF1R influences specific signaling pathways and molecular events would enhance the scientific depth of the discussion.
- Discussion of Limitations: It would be valuable for the authors to acknowledge any current limitations in our understanding of CSF1R expression and function in cancer cells. Recognizing these gaps in knowledge would provide a more nuanced perspective and enhance the overall scientific integrity of the section.
- Question Posed in the Section: The section raises several intriguing questions about CSF1R expression and downstream signaling mechanisms in cancer cells. While this is commendable, I suggest the authors provide some speculative insights or hypotheses to guide future research in these areas.
- Figure 2: The schematic representation of CSF1R signaling in different cancer cell types is a strong addition. It effectively summarizes the diverse roles of CSF1R. However, I recommend that the figure be referenced in the text to guide readers to its relevance in the context of the discussion.
Author Response
Reviewer 3
Comments and Suggestions for Authors
The manuscript provides a comprehensive overview of the current landscape of CSF1R inhibitors in cancer treatment. It effectively highlights the therapeutic potential of Pexidartinib (PLX3397) in various solid tumors and their impact on tumor growth and metastasis. The author emphasizes the need for further investigation into the molecular mechanisms regulating CSF1R expression in cancer cells and raises pertinent questions for future research. While generally well-structured and interesting to read, some comments for author consideration are listed below:
Comments for authors:
Title:
The title accurately reflects the content of the study and is concise and informative.
Introduction:
- While the section discusses CSF-1R expression in cancer cells, it could benefit from further clarification on the functional consequences of this expression. Elaborating on how CSF-1R expression in cancer cells contributes to specific aspects of tumor biology would enhance the section's depth.
Response: We thank the reviewer for the positive comments and constructive suggestions. As suggested, we integrated this part of the Introduction with the following text (lines 58-63):
“Additionally, CSF-1R expression was found in the cancer cell membrane on different tumor types, where it is associated with the main cancer hallmarks including increased proliferation, migration and drug resistance [11–14]. The functional consequences of CSF-1R expression in tumor cells make it an interesting target for anti-cancer therapy. Indeed, CSF-1R induction induced high proliferative index and anti-apoptotic signaling in breast and cervical cancer cells [11,14].”
- The addition of specific examples or case studies illustrating the impact of CSF-1R expression in different cancer types could provide a more concrete and relatable dimension to the discussion.
Response: As suggested, a new section has been added at lines 63-66, as following: “In other tumors, including mesothelioma, CSF-1R+ cells exhibited chemoresistance and expressed pluripotency-related genes [12]. Similarly, altered expression of CSF-1R was found in resistant melanoma cells with BRAF and MAPKs mutations [13].”
- In-text Citations: While the section makes several references to studies (e.g., "Src signaling correlated with CSF1R activation (52)"), I recommend the inclusion of more citations. Directly citing relevant studies would strengthen the section and provide readers with specific sources for further exploration.
Response: As suggested, more citations have been included in the text.
- Mechanistic Details: The section could benefit from a more in-depth exploration of the molecular mechanisms underlying the effects of CSF1R in cancer cells. Including detailed insights into how CSF1R influences specific signaling pathways and molecular events would enhance the scientific depth of the discussion.
Response: We agree with the reviewer that more details are needed in this section. However, while the signalling pathways induced by CSF-1R in macrophages are well characterized, the current knowledge of the molecular mechanisms of CSF-1R action in cancer is still very limited and require more in-depth exploration.
- Discussion of Limitations: It would be valuable for the authors to acknowledge any current limitations in our understanding of CSF1R expression and function in cancer cells. Recognizing these gaps in knowledge would provide a more nuanced perspective and enhance the overall scientific integrity of the section.
Response: One of the limitations is the fact that CSF1R expression in cancer cells is low as compared to that in TAMs and in other tumor-associated myeloid cells. This concept was added at lines 348-351, as following: “Main issues in the study of CSF-1R expression and function in cancer cells include the fact that it is expressed at low levels compared to the expression in myeloid cells, making it difficult to detect in some experimental settings, such as tumor tissues, where monocytes/macrophages represent the majority of CSF-1R+ cells.”
- Question Posed in the Section: The section raises several intriguing questions about CSF1R expression and downstream signalling mechanisms in cancer cells. While this is commendable, I suggest the authors provide some speculative insights or hypotheses to guide future research in these areas.
Response: As suggested, some speculative insights have been added at lines 354-356 “It is possible to hypothesize that CSF-1R is positively expressed in a specific population of cells or its expression is differently regulated during the cell’s life cycle.“ and at lines 358 “ making CSF-1R in tumor cells a negative predictor of outcome.” and at lines 362-365 “Understanding the receptor’s cellular localization may provide important indications on its activity and the mechanism regulating its function in tumor cells, as the CSF-1R nuclear localization may suggest the transcriptional activity of the receptor.”
- Figure 2: The schematic representation of CSF1R signaling in different cancer cell types is a strong addition. It effectively summarizes the diverse roles of CSF1R. However, I recommend that the figure be referenced in the text to guide readers to its relevance in the context of the discussion.
Response: As suggested, the Figure 2 was referenced several times in the text.
Reviewer 4 Report
Comments and Suggestions for Authors
In this work, authors summarized the current data available on the expression and activity of CSF-1R in different tumor types. In addition, they mentioned recent therapeutic strategies, including monoclonal antibodies and small molecules inhibitors, targeting the CSF-1R and designed to block the pro-oncogenic role of CSF-1R in cancer cell. This is interesting topic and this study can be accepted after minor revision.
Here are the points:
· “Ide et al. reported the involvement of CSF-1R in prostate cancer carcinogenesis (20)”. Authors should give more details about this study.
· 2. CSF-1R in Cancer and 2.1. CSF-1R in Cancer Cells and 2.3. CSF-1R in Cancer Cell Proliferation The same titles were given unconceivably. These captions must be merged.
· Is there a reference for Figure 1?
· The quality of Figure 2 must be improved.
· Grammatical errors should be corrected.
· The Introduction is too short that it can be expanded.
Author Response
Reviewer 4
Comments and Suggestions for Authors.
In this work, authors summarized the current data available on the expression and activity of CSF-1R in different tumor types. In addition, they mentioned recent therapeutic strategies, including monoclonal antibodies and small molecules inhibitors, targeting the CSF-1R and designed to block the pro-oncogenic role of CSF-1R in cancer cell. This is interesting topic and this study can be accepted after minor revision.
Here are the points:
- “Ide et al. reported the involvement of CSF-1R in prostate cancer carcinogenesis (20)”. Authors should give more details about this study.
Response: Accordingly, we integrated the text with more details as reported at lines 90-95, as following: “Ide et al. reported the involvement of CSF-1R in prostate cancer carcinogenesis [20]. According to the study, low level of CSF-1R and CSF-1 mRNA were detected in human prostate cancer cell lines, indicating the autocrine activation of the receptor. Further analysis indicated the expression of the receptor in prostatic intraepithelial neoplasia (PIN) and metastatic sites, suggesting a role for CSF-1R in prostate tumor development [20].”
- 2. CSF-1R in Cancer and 2.1. CSF-1R in Cancer Cells and 2.3. CSF-1R in Cancer Cell Proliferation The same titles were given unconceivably. These captions must be merged.
Response: We agree with the reviewer, the captions 2 and 2.1 have been merged.
- Is there a reference for Figure 1?
Response: the previous Figure 1 has been replaced by a new revised Figure 1 showing the CSF1R mRNA expression (log2 transformed, using a pseudo-count of 1) in cell line models from Dependency Map (DepMap) Public 23Q4 (https://depmap.org/portal/). Cell lines are grouped by cancer types defined as Oncotree Subtype. Cancer subtypes with more than 10 cell line models are shown.
- The quality of Figure 2 must be improved.
Response: Figure 2 with higher quality has been included.
- Grammatical errors should be corrected.
Response: English editing and grammar have been revised.
- The Introduction is too short that it can be expanded.
Response: as requested by the reviewer, the introduction has been expanded.